# Cross-Cultural Control–Value Mechanisms on the Detrimental Effect of Bullying on Mathematics Anxiety

**DOI:** 10.3390/bs16010003

**Published:** 2025-12-19

**Authors:** Orhan Kaplan

**Affiliations:** Department of Mathematics and Science Education, Nizip Faculty of Education, Gaziantep University, 27700 Gaziantep, Turkey; orhankaplan@gantep.edu.tr

**Keywords:** bullying, mathematics anxiety, control-value theory, cross-cultural analysis, propensity score matching

## Abstract

Although bullying behavior is widespread and has long-lasting adverse effects, the existing literature lacks strong evidence regarding the influence of bullying on students’ mathematics anxiety, as well as on their domain-specific cognitive appraisals. The purpose of this study is to estimate causal estimates of bullying/cyberbullying on mathematics anxiety, perceived control, and perceived value. Data of the eighth-grade cohort of the Trends in International Mathematics and Science Study (TIMSS) 2019 datasets (N = 46,256; 49.86% female; *M*(age) = 14.37) from Chile, Singapore, Sweden, South Africa, Turkey, and the United States were analyzed using the propensity score matching method, which yields causal inference estimates conditional on balanced covariates. The results showed that bullying victimization uniformly increased students’ mathematics anxiety across the countries. The effect of bullying victimization on students’ mathematics-related control appraisal was significant for the countries, except for Chile. This effect was more divergent on value appraisals towards mathematics. Sensitivity analyses corroborated the results. The findings suggest that cognitive appraisals may not fully capture the emotional consequences of bullying, contrary to cognitive appraisal-mediated pathways of control–value theory. Multi-country findings position bullying as an antecedent of mathematics anxiety, highlighting the need for interventions grounded in psychological, sociocultural, and educational policy factors to protect victims from its harmful effects.

## 1. Introduction

Bullying, which is a form of antisocial behavior involving violence and victimization in schools, is a worldwide issue ([33]) that negatively impacts individual mental and physical well-being, social skills, and academic outcomes. Among Organisation for Economic Co-operation and Development (OECD) countries, on average, 8% of 15-year-old students were bullied regularly ([54]). Similarly, an international study found that the frequency of experiencing bullying ranged from 5% to 70% among 11-, 13-, and 15-year-old children in Europe and North America ([20]). Among many detrimental effects, bullying victimization adversely impacts student academic performance ([21]; [47]), including mathematics achievement (e.g., [16]; [47]). It diminishes students’ sense of belonging to school, which in turn reduces their engagement and consequently their academic achievement ([16]). The developmental trajectory of bullying-related behaviors reaches its highest level in middle school ([49]), and some of the negative consequences of bullying persist into adulthood, such as risky behaviors, mental health problems, substance use, wealth problems, and disrupted social relationships (e.g., [12]; [45]; [91]).

Meta-analyses and systematic reviews have predominantly emphasized the psychological trauma associated with bullying and the corresponding intervention strategies for victims ([34]; [45]; [51]). Although numerous antecedents of mathematics anxiety have been researched and theorized, the literature lacks evidence regarding causal patterns of bullying with mathematics anxiety and cognitive appraisals. The effect of bullying types on general anxiety is established in the literature ([22]). However, a relatively lower level of correlation exists between general anxiety and mathematics anxiety, in comparison with that of test anxiety ([29]). Furthermore, within the framework of control–value theory (CVT; [61]), achievement emotions (e.g., mathematics anxiety) are considered domain-specific because they stem from situational appraisals within specific academic domains ([61]). [61] ([61]) and [64] ([64]) emphasized that trait-like emotions (e.g., general anxiety) influence academic emotions only indirectly by altering domain-specific cognitive appraisal. Consequently, according to CVT, mathematics anxiety specifically stems from math-related control and value appraisals but is indirectly influenced by general anxiety through these appraisals.

Randomized experiments provide the finest approach for establishing causality ([74]). The common use of cross-sectional data in bullying research, which introduces endogeneity issues stemming from sample selection bias and imprecise measurement of error, is mainly due to the ethical constraints of random student allocation in randomized controlled studies. Propensity score matching (PSM) can be employed in situations where randomized trials are costly, less practical, or unethical. This approach allows for the estimation of more generalizable treatment effects with balanced samples from large datasets ([26]). This approach also enables the estimation of credible evidence regarding causal relationships within PSM methodology. In summary, the acquisition of evidence concerning the effects of bullying on domain-specific anxiety and cognitive appraisals ([64]) through PSM methodology constitutes a significant contribution to CVT and the extant literature on school bullying.

Despite extensive research, bullying (including cyberbullying) victimization remains a pervasive phenomenon in middle school settings. However, there are notable gaps in the relevant literature. The examination of cross-cultural emotional development using large-scale datasets is imperative for CVT-related research, as it enhances comprehension of emotional growth in varied cultural settings ([61]). However, the majority of bullying-related studies have been conducted in individual high-income countries ([43]). Therefore, there is a scarcity of bullying-related studies with large-scale data that involve low- and middle-income countries ([9]; [23]). More importantly, the literature lacks empirical evidence regarding the effect of bullying victimization on mathematics anxiety. CVT posits that both the relationship between achievement emotions and their antecedents ([62]) and the relationship between cognitive appraisals and their antecedents are universal ([61]).

The examination of nationally representative datasets from multiple countries with diverse cultures and educational settings around the world will contribute to the contextual generalizability of the findings pertaining to the effect of a potential environmental antecedent (i.e., bullying) on mathematics anxiety and its proximal antecedents (i.e., control and value appraisals). In order to accomplish this objective, data from multiple countries with diverse levels of economic development, cultural contexts, and educational systems will be examined. The PSM analysis method will be utilized to analyze the Trends in International Mathematics and Science Study (TIMSS) data of Chile, Sweden, Singapore, South Africa, Turkey (Türkiye), and the United States. The selection of these countries was informed by the necessity to encompass a wide spectrum of cultural, socioeconomic, and educational contexts around the world, wherein the phenomenon of bullying and its ramifications on mathematics-related control and value appraisals may exhibit variability. The inclusion of these countries ensures geographical diversity, thereby augmenting the generalizability of the findings across diverse global regions.

These six educational systems exhibit varying degrees of economic development and educational stratification. For instance, Singapore and Sweden have been shown to consistently achieve top rankings in international assessments, such as TIMSS, and they allocate substantial resources per student ([46]). In contrast, Chile, South Africa, and Türkiye are grappling with pronounced socioeconomic inequalities ([24]; [40]), which may intensify both the prevalence of bullying and its academic repercussions, similar to the reported between-school relationships (e.g., [16]). Furthermore, these nations demonstrate variability in their cultural orientations toward collectivism and individualism, which may give rise to distinct patterns of peer interactions. Collectivistic societies (e.g., Singapore, Türkiye) often prioritize group cohesion, thereby potentially influencing how victims internalize bullying experiences. In contrast, more individualistic societies (e.g., United States, Sweden) may foster unique patterns of peer support and help-seeking behaviors.

A body of TIMSS-oriented research has identified country-specific trends in control–value appraisals, indicating that children in certain contexts report lower perceived control (Chile and South Africa) and value (Türkiye) in learning science ([80]). This underscores the importance of investigating the relationship between bullying and these appraisals across diverse settings. Moreover, while Türkiye and the U.S. have loosely monitored anti-bullying guidelines with uneven implementation and enforcement, other countries utilize nationally monitored and more enforceable anti-bullying programs. Specifically, Türkiye does not have anti-bullying laws (except for criminal acts), but rather the Ministry of National Education (MoNE) issues school discipline guidance and bullying awareness programs ([44]). The majority of U.S. states have adopted anti-bullying legislation, policies, reporting procedures, and investigatory processes. However, state statutes generally lack explicit enforcement mechanisms, and implementation across states is heterogeneous due to statutory language that limits liability in cases of noncompliance ([28]; [76]). Collectively, these considerations substantiate the inclusion of these six countries in order to evaluate whether the proposed mechanisms linking bullying to mathematics anxiety, control, and value appraisals are consistent across varied socioeconomic conditions, cultural orientations, and educational policy frameworks ([60]).

The findings of this study are intended to contribute to CVT research, educational psychology, mathematics education research, and school policies. Moreover, the findings of the study on the detrimental effect of bullying on mathematics anxiety are intended to stimulate constructive discourse, informed action, and the enhancement of existing initiatives aimed at reducing peer aggression and victimization, as well as mathematics anxiety.

### 1.1. Bullying/Cyberbullying

The manifestation of peer aggression, characterized as bullying, is defined as a deliberate and repetitive engagement in negative behaviors that inherently favor the perpetrator over the victim, thereby creating a power imbalance that perpetuates the cycle of abuse ([57]). Bullying can be classified into four types, namely, physical, verbal, relational/social bullying, and cyberbullying ([75]). The initial three concepts are regarded as traditional (or conventional) bullying. However, given the significant overlap in characteristics, such as repeated harassment, recent literature suggests that traditional bullying and cyberbullying should be regarded as facets of a unified construct ([42]). It is recommended that evidence on both of these types of bullying be collected together ([77]).

A multitude of factors have been identified as contributing to bullying behaviors. According to the extant literature, the incidence of bullying is higher among male students than among female students ([58]; [85]). As indicated by [16] ([16]), age functions as an additional predictor of bullying experience, thereby moderating student perception of bullying. This phenomenon occurs in conjunction with their social and emotional development. Furthermore, students with lower socioeconomic status (SES) and who are enrolled in disadvantaged schools are more likely to be victims or perpetrators of bullying ([7]; [33]). Consequently, students enrolled in disadvantaged schools with lower academic achievement and less conducive school climates—characterized by a diminished sense of school belonging—were observed to experience a higher frequency of bullying incidents ([8]; [13]; [33]). In essence, social inequalities have the potential to intensify the correlation between bullying and emotional well-being ([7]). A longitudinal study of middle school students in Korea ([73]) reported that students with low noncognitive skills were more likely to be subjected to bullying. Consequently, these demographic, socioeconomic, and school-level factors are among the most consistently reported factors influencing an individual’s likelihood of exposure to bullying.

### 1.2. Control–Value Theory and Mathematics Anxiety

The control–value theory (CVT) of achievement emotions posits that anxiety in students is a negative activating emotion arising from the interaction of control appraisals and value appraisals ([61]). Control appraisals consist of expectancies, attributions, and self-concepts of ability about achievement activities and outcomes. Value appraisals are characterized by intrinsic and extrinsic value attached to the activities and outcomes. These cognitive appraisals act as the primary sources of academic emotions or as the proximal antecedents of anxiety within the context of this study, and distal antecedents (broader factors that shape cognitive appraisals) are claimed to influence anxiety through these cognitive appraisals ([63]; [64]). Within the framework of the CVT, an individual’s anxiety is proposed to be higher in the case of low perceived control and high value of achievement outcome accompanied by a high level of uncertainty.

As a domain-specific anxiety, mathematics anxiety is defined as “thoughts and feelings about the self in relation to mathematics, such as feelings of helplessness and stress when dealing with mathematics” ([52], p. 88). Although this definition encompasses cognitive, physiological, and behavioral components, in this study, the scope of this inquiry is particularly confined to the physiological dimension of anxiety (i.e., nervousness), in which individuals elicit emotional responses to mathematical tasks and activities in school. As posited by the CVT, factors pertinent to the perception of control and value, including students’ academic self-concepts and interests, are recognized to be structured predominantly in domain-specific manners ([61]). Following these corollaries and the previous argument regarding the relation of general anxiety with mathematics anxiety, this domain-specific anxiety is treated as a separate construct.

### 1.3. Bullying in Control–Value Theory

Within the framework of the CVT ([60], [61]; [64]), higher mathematics anxiety emerges as a consequence of the interplay between an individual’s high valuation of mathematics yet low control appraisals about achieving it, which manifests itself as nervousness and worry. [64] ([64]) propose that “to the extent that control and value appraisals function as proximal antecedents, any more distal antecedents should influence achievement emotions by affecting these appraisals to begin with” (p. 123). In accordance with this preposition, if bullying exerts an influence on mathematics anxiety, an achievement emotion, its effect on math-related cognitive appraisals should also be significant.

Bullying, as a stressor, has the potential to directly influence math-related control and value appraisals as its proximal antecedent and mathematics anxiety as its distal antecedent, an environmental factor. As indicated by the findings of [32] ([32]), students who frequently encounter school bullying exhibit lower executive functioning capabilities. In the presence of induced stress or intrusive thoughts, students allocate a greater proportion of their working memory resources to concerns rather than to mathematical tasks. This allocation of resources has been shown to exacerbate anxiety levels due to a decrease in perceived control, which subsequently leads to a reduction in working memory ([5]). Therefore, the stress and intrusive thoughts that result from a history of bullying may have a detrimental impact on an individual’s mathematics anxiety, potentially exacerbating the impact of a prolonged and reinforced feedback loop. Given that many students perceive mathematical tasks as stressful, experiences of bullying and recollections of those experiences may impair students’ cognitive capabilities and relevant appraisals.

The phenomenon of bullying may have different effects on cognitive appraisals and mathematics anxiety, depending on the cultural and educational environment. In contexts such as Chile (and South Africa), where peer aggression is frequently normalized in student culture and perceived as an inherent component of the school environment ([2]), the association between bullying and cognitive appraisals may be diminished. It has been demonstrated that nations that possess legislative anti-bullying frameworks and robust welfare-oriented educational cultures—such as Sweden and Singapore—may potentially mitigate the adverse effects of bullying on students’ perceived sense of control through institutional interventions and classroom norms that impart a feeling of security and stability. In contrast, nations with fragmented anti-bullying legislation, such as the United States (and Türkiye, where guidance by the Ministry of National Education is imposed without an anti-bullying legislative framework), often demonstrate higher levels of variation in school enforcement, peer norms, and perceived safety ([76]). In these contexts, bullying can exert a more pronounced influence on students’ cognitive appraisals and emotional responses. Moreover, the effect of bullying on mathematics anxiety was found to be higher in countries with higher self-expression, such as Sweden, compared to countries with lower self-expression, such as Türkiye ([93]). The CVT posits that the factors that influence achievement emotions primarily impact cognitive appraisals ([61]; [64]). However, there is a lack of comparison regarding whether the degree of environmental factors influences control or value appraisals more strongly. A previous research study ([14]) showed that peer exclusion exerted a greater reduction in classroom participation (a manifestation of perceived control) compared to school avoidance (linked to negative value of school). However, the possible effects of factors such as the importance attributed to academic achievement and high-stakes tests, school climate, and social support structures across nations may moderate the magnitude of these paths differentially. Nevertheless, given available empirical evidence, it would be premature to draw a definitive conclusion regarding this comparison across cultures.

### 1.4. Research Aim

The purpose of this research is to estimate the change in mathematics anxiety, math-related control, and value appraisals for students who are exposed to bullying. Additionally, the study will examine these effects across various cultural and educational contexts. This study sought to answer the following research questions: (i) What is the estimated effect of bullying on eighth-grade students’ mathematics anxiety? (ii) What is the estimated effect of bullying on mathematics-related control and value appraisals in eighth-grade students? and (iii) If the effects exist, to what extent are they generalizable or consistent across countries with distinct economic, cultural, and educational contexts? Using the aforementioned theoretical framework as a foundation, the current study hypothesizes that bullying will result in an increase in victims’ mathematics anxiety and a decrease in their mathematics-related control and value appraisals. Consistent with [14]’s ([14]) findings and another study that observed a greater decrease in perceived control relative to perceived value as mathematics anxiety increased ([86]), it was hypothesized that the estimated effect of bullying on math-related perceived control will be greater than its effect on perceived value.

In the context of this study, the phenomenon of bullying victimization is a non-random selection process influenced by observable characteristics associated with the individual and their social context, encompassing familial and educational background. Therefore, the most salient socio-demographic variables in the extant literature that have the potential to influence the selection into the hypothetical treatment condition or the outcomes (mathematics-related cognitive appraisals and mathematics anxiety) were utilized through PSM to balance the samples of those who have been bullied and those who have not been bullied. The demographic variables included in the study were as follows: student gender, age, parents’ highest level of education, student home resources (i.e., the number of books at home, internet availability, and ownership of a study desk, phone, computer, or tablet), parent expectations, language spoken at home, geographic home location, classroom size, attending a disadvantaged school, and school safety.

## 2. Method

### 2.1. Research Design

In this study, the publicly available TIMSS 2019 dataset was used. This widely known international assessment measures fourth- and eighth-grade student science and mathematics achievement. It also gathers information about student background variables using questionnaires answered by students, teachers, and school administrations. A stratified two-stage cluster sampling design was used by the International Association for the Evaluation of Educational Achievement (IEA) to collect data. For the purpose of this study, eighth-grade cohort data was used to analyze the specified associations. The data consisted of six countries (Chile, Singapore, South Africa, Sweden, Türkiye, and the United States). This sampling strategy was determined by considering purposeful sampling that takes geographic, socioeconomic, and cultural diversity into account (in addition to measurement invariance results). The final analytical data—after data management and cleaning—consisted of 46,256 students from 1440 schools.

### 2.2. Instruments

Bullying victimization is operationalized as the treatment (exposure) variable. The TIMSS 2019 dataset includes a bullying item that is derived by the IEA from the Student Bullying Scale, which consists of 11 items that students responded to regarding their experience of being bullied (see Appendix A for main variables, Appendix A for covariates). According to the [79] ([79]), the Cronbach alpha coefficients for the scale were good for each country (0.87 for Chile; 0.89 for Singapore; 0.84 for South Africa; 0.89 for Sweden; 0.84 for Türkiye; and 0.90 for the U.S.). Each of these items conceptually matches—although not one-to-one—to each of 8 items of “Scale A: Being victimized” of the revised version of the highly cited [57]’ ([57]) Olweus Bully/Victim Questionnaire for Students ([36]). Therefore, its comprehensive measurement of bully victimization establishes evidence for construct validity. Moreover, this scale puts greater emphasis on bullying through digital devices, or cyberbullying ([39]). Students who were bullied about weekly or monthly were coded as the treatment group, and the remaining students were coded as the control group.

[80] ([80]) operationalized perceived control and perceived value within the context of learning science, each encompassing three distinct items. Corresponding items for the context of mathematics were adapted to construct mathematics-related perceived control and perceived value scales. Confirmatory factor analysis (CFA) provided construct validity evidence for the univariate structure of the scales for each country. The scores of scales provided acceptable to high levels of reliability, and the CFAs, which were conducted on Mplus Version 8.6, provided very good evidence for construct validity. Following satisfactory validity and reliability evidence, a composite variable was created by averaging the values of these items. Neglecting to achieve, at the very least, configural, metric, and scalar invariance for the scores of items of latent constructs could lead to any observed differences in factor structure, factor loadings, and intercepts thereof being attributed to measurement artifacts (for instance, translation discrepancies or cultural variations that may modify item interpretation, leading to individuals’ under-reporting or over-reporting) rather than signifying authentic variation ([41]; [82]). Data from these nations were analyzed in Mplus software to find evidence for measurement invariance across countries using a multi-group CFA model. While both configural and metric invariance were supported by the data, partial scalar invariance was obtained for the measurement of perceived control and perceived value across diverse nations after certain modifications in the models.

In light of the accessible and suitable data in the TIMSS 2019 dataset, the extent of this research is constrained to self-reported nervousness as an indicator of physiological symptoms of anxiety, which individuals develop in response to mathematical tasks and outcomes in school. Although complex psychological constructs are recommended to be measured with multiple-item scales, when “the construct being measured is sufficiently narrow or is unambiguous to the respondent, a single-item measure may suffice” ([88]). Previous investigations have also found that single-item emotional measures provide comparable validity and exhibit strong correlations with multi-item scales ([87]; [25]), rendering them acceptable for specific constructs such as mathematics anxiety. Therefore, the self-reported measurement of the physiological aspect of mathematics anxiety (specifically, self-reported nervousness, an indicator of physiological arousal) by a single item is justified.

In PSM analysis, alternative explanations that may influence the selection into treatment should be controlled. In order to make credible causal inferences, factors that might influence the probability of individuals being placed in the treatment group (e.g., being bullied) should be controlled to mimic the randomization process. Several covariates (demographic, contextual, and self-reported background characteristics) were included in the PSM model to reduce selection bias. Most of these variables are fixed prior to respondents’ participation in the TIMSS assessment. For PSM analyses, binary variables were created for these dichotomous variables. For *k* levels of multicategorical variables (i.e., parents’ highest level of education, number of books in home, and economically disadvantaged students), *k*-1 dummy-coded variables were created in order to reduce the dimensionality problem in PSM analysis and ensure better balance between treatment and control groups ([26]). The final score of each of the binary variables ranged from Yes (1) to No (0). Additionally, the geographic area of school location, student perception of school safety, language spoken at home, and parental expectations variables were added to the propensity score model to reduce selection bias between treatment and control groups. The classroom size variable was answered by teachers; the geographic area for school location, school disadvantage, and parental expectations variables were answered by school administrators; and other variables were answered by students.

### 2.3. Data Analysis Procedures

Before PSM analyses, outliers were removed, and missing data were handled using the MICE technique, producing five imputed datasets. The technical specification of the PSM model and its assumptions are provided in the Appendix A.

Following the satisfaction of the assumptions of the PSM model, less biased treatment effects were evaluated on finely balanced samples by utilizing each imputed dataset corresponding to each respective country using the psmatch2 command. Before executing propensity score evaluations, the intraclass correlation coefficient (ICC) is calculated by utilizing an unconditional ANOVA with random effects, as recommended by [66] ([66]). A markedly low value of ICC (ranging from 0.01 to 0.04) did not justify the implementation of multilevel modeling for the association between bullying and the respective outcomes for the dataset of the countries. Nevertheless, due to the complex survey design employed in the collection of TIMSS data and the goal of obtaining more precise treatment effect estimates, the student weight variable and the cluster variable (i.e., school ID) were utilized in the computation of propensity scores, which were subsequently applied in psmatch2 command, in order to address the complex sampling framework, nonresponse bias, and comparable covariate distributions within the matched and control cohorts. Propensity score-weighted estimates were computed for each imputed dataset, and these estimates were subsequently pooled using Rubin’s formula ([38]). To minimize bias and examine the robustness of the results, greedy matching algorithms of ‘nearest-neighbor matching with caliper’, ‘caliper matching’, and ‘kernel matching’ were performed, which were previously proven to produce similar results to optimal matching methods when the sample size is large ([6]). The literature suggests the use of nonreplacement (e.g., (71)) in nearest-neighbor matching with a caliper size of less than 0.25 standard deviation of the estimated propensity scores ([69]). Nearest-neighbor matching with a caliper of 0.1 and nonreplacement options, which provided better bias reduction compared to larger caliper sizes, were performed. Kernel matching was performed using the Epanechnikov kernel framework, which produced better covariate balance results compared to the Gaussian kernel framework. Due to kernel matching yielding the most favorable overall bias reduction subsequent to matching in comparison to nearest-neighbor and caliper matching, treatment effects employing the kernel matching method are reported. To facilitate the interpretation of the change in outcomes, a logarithmic transformation was applied to outcome variables in analyses, thereby allowing the treatment effects to signify the approximate percentage change that occurs in outcomes when students are bullied.

Finally, sensitivity analyses were performed to obtain information regarding the sensitivity—to unobserved confounding—of the effect of bullying on the constructs of interests, which is a required PSM assumption along with the common support overlap condition. Although the unconfoundedness assumption is untestable because the potential outcome for the untreated group cannot be measured, to obtain hypothetical evidence for the susceptibility of the estimations to confounding variables, sensitivity analyses employing the sensatt command ([48]) in Stata Version 14.1were performed. To accommodate the features of the sensatt command, relevant kernel matching outputs were generated using the attk command, which can be run with weight and cluster variables. This counterfactual simulation-based sensitivity analysis tests the robustness of the baseline model estimates against the simulated case that adjusts for the effect of an unobserved binary confounder *U* on outcomes and selection into treatment. The four parameters of this confounding factor are specified by $p_{ij}$, with $i,j\in \left\{0,\;1\right\},$ which “give the probability that *U* = 1 in each of the four groups defined by the treatment status and the outcome value” ([48]). The simulated parameters (p_ij_) of *U* are set at p_11_ = 0.40, p_10_ = 0.30, p_01_ = 0.30, p_00_ = 0.20, and the mean values of the outcomes are used as the threshold for creating the binary transformation of the outcomes, which is the default choice for binary transformation by sensatt. For example, the specification of p_11_ = 0.40 for the outcome mathematics anxiety assigns 40% probability that the confounder *U* is present for students being bullied (*i* = 1) and their mathematics anxiety level is above the mean (*j* = 1). Similar interpretations can be applied to other parameters. Although a substantial number of covariates were balanced through propensity score matching, *U* may correspond to other potential factors such as noncognitive factors ([73]), prior mental health issues, or environmental stressors. As the majority of these factors are influenced by the covariates, their effects on either the outcomes or selection into treatment may not be substantial for this cohort. Consequently, these moderate-to-high probability parameters are deemed reasonable for estimating the simulated effects of *U*. All statistical analyses were conducted in Stata software (Version 14.1).

## 3. Results

Propensity score analyses were performed for all countries separately. In the ‘Appendix A’, descriptive information, additional results of analyses for handling missing data (see Appendix A), reliability and validity evidence prior to creating the composite variables for perceived control and value (see Appendix A), and a comparison of treatment effects across multiple matching methods are provided (see Appendix A).

### 3.1. Descriptive Statistics

Descriptive statistics of our variables of interest were obtained from each imputed dataset using student weight as the probability weight function and the school ID as the cluster variable, and then Rubin’s rule was applied to obtain pooled parameter estimates (see Table 1).

The statistics indicate that eighth-grade students in South Africa reported the highest incidence of bullying, whereas Sweden exhibited the lowest prevalence. The trend associated with mathematics anxiety closely mirrors the trend observed in bullying within these country-level statistics. Even though the data on perceived control suggests an inverse association with bullying, the changes in perceived value indicate a somewhat close association with the trends in bullying and inversely with mathematics anxiety. Overall, mathematics anxiety is higher for countries where the perceived value of mathematics is higher than the perceived control for learning mathematics.

### 3.2. Propensity Score Analyses and Treatment Effects

As a response to our research questions, the average treatment effects on the treated (ATT) are reported in Table 2.

Propensity score analyses results indicated that the experience of being bullied significantly influences eighth-grade students’ mathematics anxiety across all countries. The treatment (exposure) effect $\left(\tau \right)$ of bullying victimization on mathematics anxiety occurs highest among Swedish students (12.5% or τ=0.125), and the change is the lowest among South African students (τ=0.058). A significant decrease in perceived control occurs in bullied students, with students from Türkiye having the highest decrease (τ=−0.108). Surprisingly, this treatment effect was not significant for Chilean students. The treatment effect of being bullied on the perceived value of mathematics was significant but small for the U.S. and Türkiye (τ=−0.020 and τ=−0.029, respectively). This effect was nonsignificant for other countries.

### 3.3. Matching Quality: Assessing Balance and Bias

The results confirmed the balance between the treatment and control groups. The common support regions, namely, the overlap between distributions of propensity scores of treatment and control groups, were satisfactory for all countries (see Appendix A).

In each performed PSM model, the percentage of mean bias on unmatched and matched samples was below 5% in post-matching, and *t*-tests that compared the mean value of covariates before and after matching were nonsignificant. Pseudo *R*^2^ values assist in interpreting the degree to which covariates explain the heterogeneity in the treatment assignment. Table 3 shows that the specified covariates explained the students’ assignment to treatment condition, or the conditional probability of being bullied, in a range from 5.1% to 7.9%. Using these specified covariates, the procedure eliminated a significant amount of bias in the incidence of being bullied in post-matching (reduced to the range of 0.1% to 0.4%). Thus, the satisfaction of the assumptions of PSM justifies the interpretation of the PSM outputs.

### 3.4. Robustness of the Results and Sensitivity Analysis

To check the robustness of the kernel matching results, the treatment effects of bullying on outcomes were compared with the estimates obtained from the ‘nearest-neighbor matching with caliper’ and ‘caliper matching’ techniques. In general, the treatment effects of bullying victimization on mathematics anxiety were robust to the matching techniques. Kernel matching and caliper matching results were very similar, and nearest-neighbor matching estimates were slightly different. Despite some exceptions with small differences, similar patterns were observed for the estimation of treatment effect on perceived control and perceived value.

To examine the possible effects of unobserved confounding variables on both treatment assignment and the resulting outcomes, sensitivity analyses on significant effects were conducted. Specifically, the simulation of the effect of an unobserved confounder *U* on the assignment of treatment and the resultant outcomes illustrated that the treatment effect retained its significance even in the face of a hypothetical confounding with moderate odds ratios of the simulated parameters of *U* (See Table 4). The inclusion of the hypothetical confounder *U* resulted in less than 10.9% reduction in the treatment effect of being bullied on mathematics anxiety. Selection effects were around 1.7, and outcome effects ranged from 1.6 to 1.7. For perceived control, due to a hypothetical variable, simulated ATT was up to 17.7% higher than the baseline ATT estimate, with selection effects ranging from 1.7 to 1.8 and outcome effects ranging from 1.6 to 1.7. Finally, for the effect of bullying on perceived value, simulated ATT differed as much as 12.3% higher than the baseline ATT estimate for Türkiye and the U.S., with selection effects of 1.7 and outcome effects of 1.6. These findings indicate that the detected treatment effects possess an acceptable degree of resilience against potential unobserved confounding and support the interpretation of the causal inferences; however, prudence is advised in the interpretation of these results, as unmeasured elements may still introduce biases that could impact the validity of the results.

## 4. Discussion

In this section, the empirical cross-cultural evidence regarding the effect of bullying victimization on eighth-grade students’ mathematics anxiety, math-related control, and value appraisals is discussed in line with their hypotheses.

In conjunction with the first hypothesis, bullying exerted a substantial influence on mathematics anxiety across all six nations. This uniform cross-cultural finding posits that bullying can be positioned as an environmental variable/distal antecedent of mathematics anxiety within the CVT. Previous studies substantiated this finding through analogous evidence by elucidating the association between an individual’s internalization of anxiety stemming from bullying victimization, which culminated in psychological difficulties and diminished academic achievement ([21]; [47]), including mathematics achievement ([16]; [47]). The increase in bullying-related mathematics anxiety may appear as an unwanted outcome. However, in conjunction with fear—since both emotional states are closely connected, anxiety should be regarded as a protective and adaptive evolutionary mechanism that increases fitness to cope with external dangers ([50]), specifically bullying within the framework of this study. Therefore, although both of these factors are detrimental to academic achievement and well-being, preventing bullying should have both moral and educational priority in this causal connection. According to the CVT ([61]), the interplay between control appraisals and value appraisals, accompanied by a certain level of uncertainty towards learning mathematics, can be claimed to be the cause of increased mathematics anxiety. However, the mediating role of cognitive appraisals on the effect of bullying victimization on mathematics anxiety is beyond the scope of this study.

The results showed that bullying decreased students’ perceived control in the countries with distinct cultural backgrounds, except for Chile. The effect of bullying on students’ perceived value was significant only in the U.S. and Türkiye. Numerous studies in developmental psychology and neuroscience underscore that bullying victimization as a manifestation of psychological distress correlates with neuromodulation and limbic dysregulation in children and impedes their emotional processing and executive functioning, including cognitive reasoning, cognitive flexibility, and learning ([59]). Moreover, stress has deleterious effects on the structure and functions of the prefrontal cortex and hippocampus and impacts an individual’s working memory, which has a key role in learning mathematics ([3]). Similarly, in the school environment, where bullying is frequent, normalized, and awarded a social status, victims of bullying were found to experience cognitive dissonance and emotional instability about whether to act upon and defend themselves ([7]; [78]). In summary, consistent with the CVT, impairments in cognitive abilities as a result of bullying, in turn, may lead to a decrease in cognitive appraisals in multiple ways. In an environment where bullying is an immediate threat or a potential future threat due to recurrence, individuals may shift long-term academic priorities away as survival concerns become a priority. This may result in lower control and value appraisals towards mathematics. As this priority consumes a considerable amount of their cognitive resources and the detrimental effect of stress (e.g., stress stemming from exposure to bullying) on the cognitive structure and functions of the brain ([3]) may exacerbate this cognitive load, an individual’s control over outcomes may diminish, along with a heightened level of uncertainty, which may explain the increase in mathematics anxiety.

The effect of bullying on value appraisals was only significant in the U.S. and Türkiye; however, these effects were small. This finding suggests that bullied students’ intrinsic value and/or extrinsic value towards mathematics (e.g., mathematics achievement is crucial for higher education and upward mobility) may generally be more resilient in Chile, Singapore, South Africa, and Sweden despite having less control over achieving these goals. Although bullying victimization is found to be an antecedent of mathematics anxiety, the path from bullying victimization to perceived value of mathematics—in addition to the path to perceived control for Chilean students—was not negatively significant for all countries, as implied by the claim of the CVT that the effect of environmental factors on emotions is fully mediated through cognitive appraisals. Multiple studies that have reported partial mediation (e.g., [15]; [92]) or no mediation (e.g., [18]) of cognitive appraisals on various emotions suggest that the generality of the CVT may be more limited than assumed. However, the current study only examined the self-reported physiological aspect of mathematics anxiety. Therefore, this contrary evidence to the CVT for some country data should be interpreted cautiously. Nevertheless, further research and theoretical elaboration are needed to provide deeper insight into the generality of the CVT.

### 4.1. Cross-Cultural Differences

[61] ([61]) highlighted the need for empirical evidence regarding the cross-cultural disparities in the development of emotions. In the present investigation, data pertaining to students from Chile, Singapore, South Africa, Sweden, Türkiye, and the United States were chosen to derive some insights into the cultural paradigms of Latin America, Asia, Africa, Europe, the Middle East, and North America, while acknowledging the existence of distinct sociopolitical and subcultural variations between countries belonging to the same cultural category. The first unexpected result was the nonsignificant effect of bullying on Chilean students’ perceived control towards mathematics. [61] ([61]) stated that
“emotions in settings that are a product of cultural evolution are thought to depend on adaptive interpretations of the situation and one’s own competencies to manage the situation…emotions related to activities and outcomes in these settings [including schools] are thought to be cognitively mediated”.(p. 10)

While small differences in treatment effects found in this study may be attributed to the individual’s interaction with and adaptation to environmental and sociocultural elements, individuals’ cognitive functioning to develop mathematics-related symptoms to bullying victimization does not vary greatly in terms of the treatment effect sizes. Additionally, the existing differences in treatment effects between countries cannot be simplistically reduced to individualist–collectivist distinction in cultures, as the findings of this study showed that both individualist and collectivist countries existed in either significant or nonsignificant treatment effect groups.

In social environments conducive to autonomy and choice, individuals experience an increase in their sense of agency and a corresponding rise in their perceived level of control over their learning ([4]). In Chile, although bullying is formally acknowledged at the policy level, bullying is normalized among students; it is widely accepted as an inherent part of school life; and students rarely report bullying incidents to teachers or parents ([2]), which may partly explain the nonsignificant effects of bullying on perceived control and value. Moreover, in Chile, those with upper secondary education earn 50% more than those without an upper secondary education, which is the highest difference among all OECD countries ([55]). Taken together, mathematics as a gatekeeper for higher levels of education occupies a significant position in Chile; therefore, victims of bullying may cultivate their sense of autonomy and structure their objectives and choices toward higher levels of proficiency in learning mathematics for upward mobility to offset reduced social standing within the proximal social milieu. Such a coping mechanism may mitigate the negative effect of increased mathematics anxiety on perceived control. Alternatively, the results showed that Chilean students, generally, had lower levels of bullying victimization, mathematics anxiety, and control and value appraisals. A high level of awareness of bullying incidents among school administrators in Chile has been documented ([56]). In accordance with the “Peaceful Coexistence Campaign,” social–emotional skills education has been incorporated into the nation’s educational agenda. The normalization of school bullying, significant disparities in students’ social and emotional skills in Chile, and insufficient training for social–emotional learning in teacher education programs ([56]), which might result in discrediting bullying as a threat to cognitive appraisals, and the influence of social desirability bias may also be potential explanations. Further empirical evidence may clarify the degree of accuracy of these propositions.

Contrary to the second hypothesis, bullying did not significantly influence Chilean, Singaporean, South African, and Swedish students’ value appraisals towards mathematics. Namely, bullying victims in these countries showed relatively more resilience to bullying incidents compared to those in the U.S. and Türkiye. This can be explained by a combination of psychological, sociocultural, and educational policy-related factors. A comparable international analysis revealed that the annual rate of change in age-standardized daily-adjusted life years (DALY) associated with anxiety disorders resulting from bullying victimization for both genders, spanning from 1990 to 2019, indicated that Türkiye and the USA exhibited significantly elevated DALY rates (each between 1.2% and less than 1.4%) in contrast to Singapore (ranging from 0.6% to less than 0.8%), South Africa (between 0.2% and less than 0.4%), and Sweden and Chile (both at 0%) ([30]). These proportions about the change in life expectancy due to anxiety disorders indicate that members of some cultures suffer more from bullying, which might be due to a lack of familial and societal support mechanisms for these victims. Furthermore, according to an OECD report ([1]), the percentages of disadvantaged 15-year-old students who are resilient (i.e., restricted to ‘obtaining higher achievement despite their SES disadvantage’) are 43.4% in Singapore and 25% in Sweden, which are significantly higher than the U.S. (22.3%) and Türkiye (7.2%). However, this report does not provide an explanation for the Chilean students’ observed low resilience (7.2%). Alternatively, the well-being of students may serve as a mitigating factor against the adverse impacts of bullying on the perceived value of mathematics among students. Adolescents in the U.S. and Türkiye expressed lower levels of life satisfaction (54.8% and 38.0%, respectively) in comparison to their counterparts in Chile (57.6%) and Sweden (60.2%) ([53]). Student well-being was found to be positively associated with parent and teacher support, which are positively linked to the students’ perceived value of mathematics ([89]). While these statistics offer justification for the significant decrease in value appraisals towards mathematics within the U.S. and Türkiye, a complex interplay of multiple determinants may explain the resilience of eighth-grade students in other countries to mitigate the decline in the valuation of mathematics.

Social capital in the context of valuing mathematics may also provide a plausible alternative explanation. A study found that while the mathematics achievement of students in the United States was predominantly accounted for by the resources available at home and school, students in Singapore primarily relied more on their attitudes, motivation, and expectations to achieve higher academic outcomes ([35]). Furthermore, students in more disadvantaged schools tended to react to bullying incidents less ([7]), which might explain the relatively lower detrimental effect of bullying on students’ control and value appraisals in South Africa, which has a lower gross domestic product (GDP) per capita. High parental value of mathematics is associated with students’ value appraisals ([10]), and bullied students with strong familial connections have higher levels of resilience as a result of better psychosocial behavioral and emotional adjustment ([72]). Mathematics is generally crucial for social mobility in Asian nations, Africa, and Latin America, and its high valuation in these cultures may also mitigate the effects of bullying, as these cultures also emphasize strong family connections. Thus, in environments with supportive social frameworks and high esteem for mathematics, students may show increased resilience to bullying’s negative impact on their perception of the value of learning mathematics.

School anti-bullying policies and protective national laws are an essential component of an integrated approach to addressing the phenomena of school violence and bullying. National initiatives aimed at combating bullying can reduce its harmful effects on students through preventive and supportive actions by school administrations. According to the United Nations Educational, Scientific and Cultural Organization ([81]), several actions are taken against bullying and violence by the countries in this study: 2011 the School Violence in the General Education Law and the Peaceful Coexistence Campaign in Chile, Teachers Guidelines [and 2011 the Protection from Harassment Act 17] in South Africa, 2009 the Discrimination Act and 2010 the Skollagen (Education Act) in Sweden, and the Protection from Harassment Act in Singapore have instituted specific legislative frameworks aimed at curbing bullying behavior. However, there is no explicit legislation addressing or preventing school bullying in Türkiye; the Ministry of National Education’s approach towards combating this issue has been gradually developing and predominantly consists of regulatory action plans tailored for educational institutions. Furthermore, while the Improving America Schools Act and the Safe and Drug-Free Schools and Communities Act establish a general foundational framework in the U.S., a singular federal statute specifically addressing school bullying remains absent ([81]). Instead, individual states have assumed the responsibility to initiate the legalization and implementation of anti-bullying programs. These policies highlight that the inexplicit and ubiquitously less enforceable educational policies of the U.S. and Türkiye may also explain their divergent outcomes compared to other countries’ legislative educational policies.

In summary, the cited literature implies that psychological, sociocultural, economic, and state-level educational policy-related factors concurrently provide insights into the effect of bullying on control and value appraisals.

### 4.2. Limitations and Future Recommendations

In addition to previous suggestions, in this study, only the self-reported physiological aspect of mathematics anxiety was examined. The nonsignificant effects of bullying victimization on cognitive appraisals in some countries were discussed with some relevant factors. These unexpected findings suggest that the negative effects of bullying on mathematics anxiety are mostly direct, or the effect of bullying is not uniquely transmitted through cognitive processes. Cross-cultural investigation of the effect of bullying victimization on the cognitive (e.g., perception of task difficulty and negative self-image) and behavioral aspects of mathematics anxiety (e.g., task avoidance, lower engagement within classroom, and less peer interaction) through cognitive appraisals can provide more information. Propensity score analysis mimics a randomized controlled trial and provides robust results when its assumptions are met. Many relevant covariates were taken into account, and sensitivity analysis did not reveal a large change in the treatment effects in cases of a hypothetical confounder, which suggested that the observed treatment effects were not solely an artifact of such bias. However, the constraint of being unable to comprehensively account for all potential confounders (or potential mediators such as executive functions, emotion regulation capacity, peer relationships, and support) highlights the necessity for the exploration of alternative explanations for the link between bullying and mathematics anxiety and that of control and value appraisals, as implied in causal inference theory ([71]).

Cross-cultural analysis results were elucidated by some psychological, economic, and sociocultural factors and state-level educational policy. However, other factors might explain this pattern. Additionally, the literature has a shortage of cross-cultural explanations pertaining to understanding the etiology behind emotional diversity. Cross-national studies within cultural groups would provide the generalizability of the findings across countries that belong to the same cultural group. The persistence of bullying prior to eighth grade may have influenced unobserved cognitive and non-cognitive outcomes, i.e., the joint causality problem ([27]), such as mathematics anxiety. Due to the limited number of variables in the TIMSS dataset, this study did not control for individuals’ previous bullying experience and its cumulative effects through cognitive and non-cognitive variables. Future research can partial out the longitudinal effects of these relations. Finally, self-report and social desirability bias are potential threats to internal validity. Differential classification of bullying, such as over-reporting by anxious students or under-reporting by students who normalize bullying, might have biased the treatment effects. The incorporation of other sources of data into future research can eliminate the possibility of such bias.

## 5. Conclusions

Peer bullying is a global problem that has detrimental effects on emotion and cognition. It increases mathematics anxiety and decreases mathematics-related control and value appraisals in most countries, and this harmful effect is not fully cognitively mediated. However, the cross-national analysis provides insights into mitigating the detrimental effects of bullying on students’ control and value appraisals. Further research is necessary for clarifying the implications and potential explanations of the findings of this study within the CVT framework. Although attempts to reduce anxiety, and mathematics anxiety in particular, are necessary for the health and overall well-being of victims, in this context, bullying-induced mathematics anxiety should be viewed as a defensive physiological symptom of peer aggression rather than the primary problem itself. While the establishment of a legal framework for the prevention of peer bullying may be a crucial element, as discussed in the cross-cultural analysis, it is essential that the psychological resilience and well-being of victims be developed through the enhancement of individual autonomy and support mechanisms provided by family, teacher, school, and the proximal social environment. In the literature, many anti-bullying initiatives have failed to substantially eliminate bullying in schools, with the same prevalence levels returning after the program was discontinued. Although bullying is a harmful antisocial behavior, it is necessary to recognize the evolutionary functions of bullying that are historically embedded in some human social contexts, which demands intervention through education and social learning to replace it with healthy social relationships. Future anti-bullying programs that comprehensively target these cognitively mediated functions of bullying (i.e., reputation, resources, deterrence, recreation, and reproduction) ([84]) may yield more promising results. Further research, data-driven interventions, and more policy actions are needed to eliminate this harmful behavior from pupils’ social lives.

## Figures and Tables

**Table 1 behavsci-16-00003-t001:** Pooled country-level descriptive statistics.

Country	N_student_	N_school_	Bullying	Mathematics Anxiety	Perceived Control	Perceived Value
Chile	4108	164	1.304 (0.009)	2.321 (0.016)	2.550 (0.014)	2.544 (0.013)
Singapore	4843	153	1.404 (0.009)	2.624 (0.014)	2.616 (0.013)	2.806 (0.011)
South Africa	20,637	519	1.827 (0.006)	2.737 (0.009)	2.326 (0.007)	3.168 (0.007)
Sweden	3977	150	1.239 (0.009)	2.097 (0.017)	2.691 (0.016)	2.398 (0.014)
Türkiye	4030	181	1.317 (0.009)	2.575 (0.018)	2.546 (0.016)	2.892 (0.013)
U.S.	8661	273	1.359 (0.007)	2.227 (0.014)	2.678 (0.013)	2.627 (0.012)

Note. Linearized standard errors (values in parentheses) were derived employing the ‘svy’ function in Stata that allows the fitting of statistical models for complex survey data through the adjustment of results for the probability weight and school ID as the cluster variable.

**Table 2 behavsci-16-00003-t002:** Results from kernel matching method: ATT of those being bullied.

Country	Mathematics Anxiety	Perceived Control	Perceived Value
Chile	0.075 *** (0.017)	−0.023 (0.014)	0.008 (0.014)
Singapore	0.077 *** (0.014)	−0.040 ** (0.014)	−0.005 (0.011)
South Africa	0.058 *** (0.007)	−0.030 *** (0.006)	0.001 (0.005)
Sweden	0.125 *** (0.020)	−0.048 ** (0.017)	−0.020 (0.016)
Türkiye	0.101 *** (0.019)	−0.108 *** (0.016)	−0.029 * (0.014)
USA	0.091 *** (0.012)	−0.061 *** (0.011)	−0.020 * (0.010)

Note. Pooled values are proportional changes in the outcome (e.g., 0.075 ≅ 7.5%). Standard errors are given in parentheses. * *p* < 0.05; ** *p* < 0.01; *** *p* < 0.001.

**Table 3 behavsci-16-00003-t003:** Bias reduction: the variance in incidence of being bullied that can be accounted for by the covariates.

Countries	Unmatched	Matched
Chile	0.059	0.002
Singapore	0.068	0.001
South Africa	0.051	0.002
Sweden	0.076	0.003
Türkiye	0.079	0.004
United States	0.056	0.002

Note. These pseudo R-squared values were derived from the probit estimation using psmatch2 kernel matching concerning the conditional probability of being bullied contingent on the covariates.

**Table 4 behavsci-16-00003-t004:** Sensitivity analyses for PSM estimates of the effects of bullying victimization on outcome variables.

Variables		Chile	Singapore	South Africa	Sweden	Türkiye	U.S.
Math anxiety	ΔATT (%)	10.941	9.350	12.280	6.299	9.935	9.701
	Γ	1.738	1.733	1.729	1.728	1.735	1.729
	Λ	1.719	1.662	1.634	1.660	1.693	1.637
Perceived control	ΔATT (%)	−17.697	−12.571	−12.50	−8.043	−4.358	−6.355
	Γ	1.745	1.741	1.741	1.746	1.747	1.724
	Λ	1.625	1.591	1.584	1.603	1.579	1.589
Perceived value	ΔATT (%)	-	-	-	-	−11.964	−12.349
	Γ	-	-	-	-	1.748	1.727
	Λ	-	-	-	-	1.586	1.594

Note. (i) Reported values are pooled sensitivity analysis estimates with 500 simulated replications. (ii) $\Delta \mathrm{ATT}=100\times ({\hat{\tau }}_{\mathrm{baseline}}-{\hat{\tau }}_{\mathrm{simulated}})/{\hat{\tau }}_{\mathrm{baseline}}$; Γ: Selection effect; Λ: Outcome effect.

## Data Availability

A publicly available international TIMSS 2019 data was used, which can be accessed through: https://timss2019.org/international-database/ (accessed on 20 November 2025).

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
