# Peer review of "Cross-Cultural Control–Value Mechanisms on the Detrimental Effect of Bullying on Mathematics Anxiety"

_behavsci, 2025, doi:10.3390/bs16010003_

Round 1
Reviewer 1 Report
Comments and Suggestions for Authors
Review MDPI Article
Thank you for the opportunity to review this interesting and timely manuscript. The study addresses an important topic. My comments are intended to support the authors in enhancing the clarity, methodological transparency, and the overall impact of the work.
Abstract:
Here, it is standard practise to include information about the sample, for example the total number of participants (N = ?), the breakdown by gender and the mean age (M =?).
Introduction:
Line 31: Bullying does not merely relate to school violence but you could be specific by stating that this includes violence and victimisation in schools.
Line 35: OECD is an acronym for Organisation for Economic Co-operation and Development.
Line 36: Specify what you mean by “young students”, e.g., middle school students?
Line 39: Here, I would use the term “adversely impacts” instead of decreases academic performance.
Line 41-42: I suggest rewording this sentence to: “It diminishes students’ sense of belonging to school, which in turn reduces their engagement and consequently their academic achievement.”
Line 45: Specify what you mean by “low occupational status”, i.e., less prestigious or lower-paying, low-skilled?
Line 48: The authors should add references to the literature here.
Lines 48-50: Although I appreciate that bullying and victimisation may increase students’ anxiety levels thus negatively impacting academic performance in general, I am struggling to understand the connection to mathematics anxiety. My concern is that based on what the authors have written thus far, a general link between bullying and overall anxiety does not automatically justify a specific connection to mathematics anxiety unless one argues that bullying increases overall anxiety which can spill over into performance in specific academic domains including mathematics. The wording needs to be revised here.
Lines 51-55: This sentence needs restructuring as it is too long. It would be preferable to break it down into shorter sentences.
Line 64: I do not believe this statement accurately reflects the current state of research on the topic of bullying victimisation in middle school settings. On review, bullying (including cyberbullying) prevalence in middle school has been studied extensively but perhaps important gaps remain understudied, there are methodological challenges or there are still important open questions.
Lines 74-79: There is much repetition here which makes it difficult for the reader to grasp. The authors need to be more concise.
Lines 86-90: In my opinion, the authors do not need to add the continents in brackets. Rather, name only the specific countries which were selected for participation in the study such as South Africa, Singapore, Sweden, Türkiye….etc.
Line 96: Pay attention to structure. This section would benefit by adding a new paragraph. Revise your manuscript and restructure accordingly.
Line 102: Similarly, more structure here.
Lines 107-109: Please support your claim here with references to the literature. This is a broad statement relating to a federal system such as the USA. As far as I am aware, almost all states in the USA have some kind of bullying law. However, many of these laws lack strong enforcement mechanisms. Schools are often required to have policies, but there is little legal recourse for students if schools don't follow through.
Lines 132-135: Belongs in the method section.
Lines 138-150: This section needs to be revised. In general, a comprehensive language edit would improve the precision and clarity of expression throughout the manuscript.
Lines 153-172: This paragraph is densely written and mixes multiple theoretical ideas at once, making it hard to follow. It needs revision and could be condensed into a few sentences which define and briefly convey the main ideas.
Lines 179-189: Section needs revision and restructuring. Again, the main ideas presented here could be expressed more concisely in two or three sentences.
Lines 192-221: Again, this section needs to be revised because the way it is written, it is much more complicated than it needs to be. The logic connecting bullying to mathematics anxiety to control-value appraisals is not clearly explained, even though it could be justified with clearer reasoning.
If lines 218–221 constitute the proposed hypotheses, these should be reformulated, clearly stated, and relocated into their own section. Importantly, the wording of each hypothesis requires careful attention to ensure they are precise, unambiguous, and transparent to the reader.
Line 223: Section 1.4 Research Aim: This section needs to focus on the aim of the study, the research questions and the rationale behind them supported by references to the literature. A new section should follow which outlines the hypotheses drawn from the research questions. I read one reference to what appears to be an overarching hypothesis in the previous section but it would be preferable to dedicate a full subsection to the hypotheses to be examined in this study. If lines 218–221 constitute the proposed hypotheses, these should be reformulated, clearly stated, and relocated into their own section. Importantly, the wording of each hypothesis requires careful attention to ensure they are precise, unambiguous, and transparent to the reader.
Method
Line 256: here adding (N = 46,256) satisfies APA formatting guidelines.
Line 267: Here, the authors briefly refer to the psychometric properties of the bullying instrument. However, there is no information about reliability and no statistical values such as Cronbach’s alpha (α): ≥ .70 to support claims. A reliability analysis could be carried out of the sample data to verify how robust the psychometric properties of the instrument are.
Line 272: Table 1 is confusing because it does not actually contain any statistically descriptive information. Usually, the scale used (e.g., Likert 5-point) and a sample item are provided in the description of the instrument. Table 1 could be converted to a figure and appended to the manuscript or added to the supplementary information.
Lines 294-301: The authors previously refer to the inclusion of physiological data pertaining to anxiety. However, I cannot find or it is not clear in the manuscript, what method was used to capture physiological data. In lines 295 and 300, there is a reference to an item from a self-report measure. This does not equate to the measurement of physiological data which uses objective measurements of biological or bodily processes, captured using sensors, medical devices, or specialised equipment measuring skin conductance or heart rate variability, for example. What is actually referred to here is a self-reported symptom of mathematics-related anxiety, so subjective.
Lines 302-303: Perhaps it would be more accurate to state here that the variables included in the PSM model are primarily demographic, contextual, and self-reported background characteristics, used as covariates to reduce selection bias?
Line 318: Section 2.3 Data Analysis Procedures: While this section is very detailed and demonstrates careful application of the statistical techniques used, the technical detail is perhaps more extensive than necessary, particularly for readers unfamiliar with the propensity score analysis approach. It could reasonably be moved to an appendix or to the supplementary material.
Results
Line 469: Table 3 needs some work. In APA 7 format, the table number comes first and is bolded. The table title comes next, is italicised and is placed on a new line below the table number. Major words are capitalised and the full stop is omitted. All tables should be formatted in this way unless the journal specifies otherwise. The country column also needs a caption to this effect.
Line 475: I am wondering if it might be better to refer to bullying victimisation as the treatment (exposure) effect here, as I feel for readers not familiar with PSM, this is confusing. Also, in line 260 in the method section, perhaps it might be better to state that: “Bullying victimisation is operationalised as the treatment (exposure) variable”.
Line 490: In APA 7 format, the figure number is placed in bold above the figure itself. The title is placed on the line below the figure number and is italicised. Major words are capitalised and the full stop is omitted.
Discussion
Line 555: Here the authors mention recommendations for future research. This should be added to section 4.2.
Lines 568-575: Although the author correctly discusses how bullying can impair executive functioning, particularly working memory, which is fundamental for mathematical learning and numerical processing, this highlights a central limitation of the study, in my opinion. None of these cognitive constructs were actually measured. As a result, while the theoretical explanation linking bullying to mathematics anxiety is conceptually sound and supported by prior research, the present study lacks the critical empirical variables needed to substantiate this. Consequently, the interpretation of how bullying leads to increased mathematics anxiety appears speculative rather than directly evidenced by the data analysed. The authors have alluded to this briefly in the limitations section (lines 725-728) but not in specific detail.
Line 573: For clarity and comprehension, the authors could improve this sentence by adding that the stress they refer to arises from exposure to bullying.
Lines 601-602: Again, the study does not appear to have included the measurement of any physical or physiological aspect of mathematics anxiety. Rather a self-report questionnaire was the chosen instrument measuring this construct.
Lines 630-632: I believe this statement does not accurately reflect the current research findings on the status of bullying in Chilean schools / education system. According to national survey data from the Chilean ministry of education which measures bullying explicitly, Chile acknowledges bullying as a real and serious problem at the institutional, research and policy levels. Its denial is not systemic, rather it occurs more at the level of students’ perceptions or peer culture. It may be that certain behaviours which constitute bullying in some countries or regions may be minimised or not labelled bullying in others and therefore, they are under-reported rather than denied.
Lines 643-644: I believe this to be somewhat of a blanket statement which does not accurately reflect the findings of the referenced report. According to this report, there is evidence that social and emotional skills education is on Chile’s agenda, that some teachers feel prepared and willing to teach these skills, and that some schools may already incorporate these practices. In fact, the referenced report provides an example of a targeted programme (Aulas en Paz: Peace education meets social and emotional skills in Chile, Colombia, p. 82-83) “that seeks to promote peaceful coexistence and reduce aggression in school environments, especially in vulnerable communities with histories of violence and substance abuse”. The authors even refer to this programme in lines 699-700. What the findings of the report do show is that there are large discrepancies in how these skills are promoted and in students’ own socio-emotional skills. Furthermore, the DOI of the referenced report listed in the bibliography list appears to be incorrect.
Lines 703-706: A statement such as this requires a reference to the source of this information, otherwise it appears baseless.
Lines 770-773: This statement as written is conceptually risky and oversimplified, possibly even inappropriate, especially in a scientific report in the context of bullying, anxiety and educational outcomes. Bullying is not generally accepted in the scientific community as a genetic predisposition. Rather, as the authors previously state, peer dynamics, school climate, family factors and individual traits play a more significant role. The word “selfish” is also inappropriate. “Harmful” is perhaps a better choice here. As mentioned previously, the manuscript could benefit from professional language editing in order to avoid making unintentionally risky statements and enhance its impact.
Lines 776-778: Although I agree with this statement, it contradicts the previously mentioned concept of bullying as genetically embedded. If bullying were primarily rooted in a genetic predisposition, eliminating or substantially reducing the behaviour through educational interventions would be extremely difficult.
In conclusion, this manuscript addresses an important and timely topic and demonstrates commendable effort in both its theoretical framing and analytical approach. However, in my opinion, several areas require substantial revision before the work can be deemed publishable. The clarity and organisation of the introduction need improvement. Key constructs such as executive functioning and physiological aspects of anxiety are discussed but not measured, and several interpretations extend beyond what the data can actually support. Additionally, some claims, particularly those relating to the nature of bullying, need to be reconsidered to ensure they align with current empirical evidence and avoid conceptual overreach. With clearer theoretical grounding, more precise methodological reporting, and careful refinement of the discussion, the manuscript could make a valuable contribution to the literature on bullying and mathematics anxiety.
Comments on the Quality of English LanguageAs mentioned above, the manuscript could benefit from professional language editing in order to avoid making unintentionally risky statements and to enhance the impact of the research.
Author Response
Comment 1:
Abstract: Here, it is standard practise to include information about the sample, for example the total number of participants (N = ?), the breakdown by gender and the mean age (M =?).
Response: Thank you for the suggestion. Each of these statistics were added to the abstract.
Comment 2:
Line 31: Bullying does not merely relate to school violence but you could be specific by stating that this includes violence and victimisation in schools.
Response: I believe you refer to the Line 28 (in the version I downloaded from the journal dashboard), it was rephrased as “antisocial behavior involving violence and victimization in schools”
Thank you for helping me clarify the statement.
Comment 3:
Line 35: OECD is an acronym for Organisation for Economic Co-operation and Development.
Response: The typo was corrected. Thank you for pointing that out.
Comment 4:
Line 36: Specify what you mean by “young students”, e.g., middle school students?
Response: It was a PISA finding derived from 15-year-old students, mostly 9th grade students. It was corrected as “15-year-old students”
Comment 5:
Line 39: Here, I would use the term “adversely impacts” instead of decreases academic performance.
Response: The term changed to ‘adversely impacts’. Thank you for the recommendation!
Comment 6:
Line 41-42: I suggest rewording this sentence to: “It diminishes students’ sense of belonging to school, which in turn reduces their engagement and consequently their academic achievement.”
Response: Thank you for the suggestion. It was corrected.
Comment 7:
Line 45: Specify what you mean by “low occupational status”, i.e., less prestigious or lower-paying, low-skilled?
Response: It was restated as “wealth problems and disrupted social relationships”, and "later violence” was rephrased as later “risky behaviors” for specificity.
Comment 8:
Line 48: The authors should add references to the literature here.
Response: Thank you for pointing that out. The statement was supported by relevant literature.
Comment 9:
Lines 48-50: Although I appreciate that bullying and victimisation may increase students’ anxiety levels thus negatively impacting academic performance in general, I am struggling to understand the connection to mathematics anxiety. My concern is that based on what the authors have written thus far, a general link between bullying and overall anxiety does not automatically justify a specific connection to mathematics anxiety unless one argues that bullying increases overall anxiety which can spill over into performance in specific academic domains including mathematics. The wording needs to be revised here.
Response: Thank you for the recommendation. The relationship between general anxiety and mathematics anxiety was discussed and the arguments were supported by relevant literature (Line 49-65).
Comment 10:
Lines 51-55: This sentence needs restructuring as it is too long. It would be preferable to break it down into shorter sentences.
Response: It was split into two sentences (Lines 67-71).
Comment 11:
Line 64: I do not believe this statement accurately reflects the current state of research on the topic of bullying victimisation in middle school settings. On review, bullying (including cyberbullying) prevalence in middle school has been studied extensively but perhaps important gaps remain understudied, there are methodological challenges or there are still important open questions.
Response: Thank you for pointing that out. The outdated citation (and its claim) was removed and the sentence was rephrased as “Despite extensive research, bullying (including cyberbullying) victimization remains a pervasive phenomenon in middle school settings. Yet, important gaps remain understudied.”
Comment 12:
Lines 74-79: There is much repetition here which makes it difficult for the reader to grasp. The authors need to be more concise.
Response: The phrase ‘and the contextual generalizability of CVT’ was removed
Comment 13:
Lines 86-90: In my opinion, the authors do not need to add the continents in brackets. Rather, name only the specific countries which were selected for participation in the study such as South Africa, Singapore, Sweden, Türkiye….etc.
Response: Thank you for the suggestion. Only the country names were kept, and the repetition of the country names in the next sentence was deleted.
Comment 14:
Line 96: Pay attention to structure. This section would benefit by adding a new paragraph. Revise your manuscript and restructure accordingly.
Response: Thank you for pointing that out. The paragraph was split into three paragraphs.
Comment 15:
Line 102: Similarly, more structure here.
Response: The paragraph was split into three paragraphs.
Comment 16:
Lines 107-109: Please support your claim here with references to the literature. This is a broad statement relating to a federal system such as the USA. As far as I am aware, almost all states in the USA have some kind of bullying law. However, many of these laws lack strong enforcement mechanisms. Schools are often required to have policies, but there is little legal recourse for students if schools don't follow through.
Response: The statements regarding Turkey and the U.S. were elaborated with relevant citations. Thank you for pointing that out.
Comment 17:
Lines 132-135: Belongs in the method section.
Response: The statements ‘In PSM analysis, alternative explanations that may influence the selection into treatment should be controlled. In other words, to make credible causal inferences, factors that might influence the probability of individuals being placed in the treatment group (e.g., being bullied) should be controlled to mimic the randomization process.’ was moved to the end of Instruments section.
Comment 18:
Lines 138-150: This section needs to be revised. In general, a comprehensive language edit would improve the precision and clarity of expression throughout the manuscript.
Response: We thank the reviewer for this observation. In response, we have revised the section to improve clarity and precision. Additionally, we conducted a language edit throughout the manuscript to enhance readability and ensure more precise expression.
Comment 19:
Lines 153-172: This paragraph is densely written and mixes multiple theoretical ideas at once, making it hard to follow. It needs revision and could be condensed into a few sentences which define and briefly convey the main ideas.
Response: Thank you for your valuable feedback. We agree that, since our study is grounded in Control-Value Theory, which conceptualizes anxiety as an achievement-related emotion, the inclusion of Russell’s definition of emotion from the affect perspective is not directly aligned with our theoretical framework. Accordingly, Russell’s (2003), Daniels et al. (2009) and Schrerer (2000) definitions from the section were removed, the section was shortened, leaving only arguments and discussion directly related to Control-Value Theory to ensure conceptual consistency.
Comment 20:
Lines 179-189: Section needs revision and restructuring. Again, the main ideas presented here could be expressed more concisely in two or three sentences.
Response: Thank you for the suggestion. As the distinction by Hembree between general anxiety and mathematics anxiety was already mentioned in the Introduction (as a correction to your earlier comment), they have been removed from the this section to ensure a more concise and consistent argument.
Comment 21:
Lines 192-221: Again, this section needs to be revised because the way it is written, it is much more complicated than it needs to be. The logic connecting bullying to mathematics anxiety to control-value appraisals is not clearly explained, even though it could be justified with clearer reasoning.
Response: We thank the reviewer for this helpful comment. In response, we have revised the section ‘1.3. Bullying in Control-Value Theory’ to simplify the writing and clarify the logic linking bullying to mathematics anxiety and control&value appraisals. The revised section now presents the reasoning in a more direct and coherent manner, ensuring that the conceptual connections are easier to follow.
Comment 22:
If lines 218–221 constitute the proposed hypotheses, these should be reformulated, clearly stated, and relocated into their own section. Importantly, the wording of each hypothesis requires careful attention to ensure they are precise, unambiguous, and transparent to the reader.
Response: We thank the reviewer for this suggestion. It was moved to the section ‘Research Aim’
Comment 23:
Line 223: Section 1.4 Research Aim: This section needs to focus on the aim of the study, the research questions and the rationale behind them supported by references to the literature. A new section should follow which outlines the hypotheses drawn from the research questions. I read one reference to what appears to be an overarching hypothesis in the previous section but it would be preferable to dedicate a full subsection to the hypotheses to be examined in this study. If lines 218–221 constitute the proposed hypotheses, these should be reformulated, clearly stated, and relocated into their own section. Importantly, the wording of each hypothesis requires careful attention to ensure they are precise, unambiguous, and transparent to the reader.
Response: We thank the reviewer for this valuable guidance. We have revised the Research Aim section to clearly articulate and distinguish the study’s aim, research questions, and their rationale with supporting literature. We also created a separate subsection dedicated to the hypotheses, as recommended. The hypotheses have been reformulated for greater precision and clarity.
Comment 24:
Line 256: here adding (N = 46,256) satisfies APA formatting guidelines.
Response: The statement and statistics about the initial data was removed.
Comment 25:
Line 267: Here, the authors briefly refer to the psychometric properties of the bullying instrument. However, there is no information about reliability and no statistical values such as Cronbach’s alpha (α): ≥ .70 to support claims. A reliability analysis could be carried out of the sample data to verify how robust the psychometric properties of the instrument are.
Response: We appreciate the reviewer’s attention to the methodological precision. Bullying item was a composite categorical item, composed of 11 items by TIMSS & PIRLS International Study Center. Cronbach’s alpha coefficients of the scale for each country was reported in the Instrument section and the citation (with the link) is provided.
Comment 26:
Line 272: Table 1 is confusing because it does not actually contain any statistically descriptive information. Usually, the scale used (e.g., Likert 5-point) and a sample item are provided in the description of the instrument. Table 1 could be converted to a figure and appended to the manuscript or added to the supplementary information.
Response: It was added to Supplementary File
Comment 27:
Lines 294-301: The authors previously refer to the inclusion of physiological data pertaining to anxiety. However, I cannot find or it is not clear in the manuscript, what method was used to capture physiological data. In lines 295 and 300, there is a reference to an item from a self-report measure. This does not equate to the measurement of physiological data which uses objective measurements of biological or bodily processes, captured using sensors, medical devices, or specialised equipment measuring skin conductance or heart rate variability, for example. What is actually referred to here is a self-reported symptom of mathematics-related anxiety, so subjective.
Response: We appreciate the reviewer’s careful attention to this point. We agree that self-reported items in TIMSS do not constitute objective physiological measurements. The phrases previously used in the manuscript may have unintentionally implied the use of physiological instrumentation, which was not the case. TIMSS collects self-reported indicators of anxiety.
Additionally, we would like to emphasize that the aim of this study is to provide a broad, cross-national perspective on students’ mathematics anxiety using a large-scale international dataset. While TIMSS does not offer objective physiological indicators, it allows us to capture population-level patterns and compare countries in a way that is not feasible with small-scale physiological studies.
To avoid any misunderstanding, we have revised the terminology throughout the manuscript to explicitly refer to these variables as “self-reported physiological symptoms/aspects of mathematics-related anxiety.” We thank the reviewer for helping us improve the precision of our wording, and the manuscript has been updated accordingly.
Comment 28:
Lines 302-303: Perhaps it would be more accurate to state here that the variables included in the PSM model are primarily demographic, contextual, and self-reported background characteristics, used as covariates to reduce selection bias?
Response: The statement was rephrased as: “Several covariates (demographic, contextual and self-reported background characteristics) were included in the PSM model to reduce selection bias.”
Comment 29:
Line 318: Section 2.3 Data Analysis Procedures: While this section is very detailed and demonstrates careful application of the statistical techniques used, the technical detail is perhaps more extensive than necessary, particularly for readers unfamiliar with the propensity score analysis approach. It could reasonably be moved to an appendix or to the supplementary material.
Response: Technical formulation of PSM and its assumptions were moved to Supplementary File. However, commonly reported data analysis procedures were kept in the section.
Comment 30:
Line 469: Table 3 needs some work. In APA 7 format, the table number comes first and is bolded. The table title comes next, is italicised and is placed on a new line below the table number. Major words are capitalised and the full stop is omitted. All tables should be formatted in this way unless the journal specifies otherwise. The country column also needs a caption to this effect.
Response: The tables were prepared according to the journal’s template. A caption was added to the country column.
Comment 31:
Line 475: I am wondering if it might be better to refer to bullying victimisation as the treatment (exposure) effect here, as I feel for readers not familiar with PSM, this is confusing. Also, in line 260 in the method section, perhaps it might be better to state that: “Bullying victimisation is operationalised as the treatment (exposure) variable”.
Response: In the ‘Instruments’ section the term was written as ‘treatment (exposure) variable’ and in the Results section ‘treatment (exposure) effect’ was introduced to the readers.
Comment 32:
Line 490: In APA 7 format, the figure number is placed in bold above the figure itself. The title is placed on the line below the figure number and is italicised. Major words are capitalised and the full stop is omitted.
Response: The figure was prepared according to the journal’s template. It was moved to Supplementary File based on your earlier comment.
Comment 33:
Line 555: Here the authors mention recommendations for future research. This should be added to section 4.2.
Response: A similar statement was added to Conclusion section. Therefore, the sentence was rephrased as “Therefore, although both of these factors are detrimental to academic achievement, preventing bullying should have priority in this causal connection.”
Comment 34:
Lines 568-575: Although the author correctly discusses how bullying can impair executive functioning, particularly working memory, which is fundamental for mathematical learning and numerical processing, this highlights a central limitation of the study, in my opinion. None of these cognitive constructs were actually measured. As a result, while the theoretical explanation linking bullying to mathematics anxiety is conceptually sound and supported by prior research, the present study lacks the critical empirical variables needed to substantiate this. Consequently, the interpretation of how bullying leads to increased mathematics anxiety appears speculative rather than directly evidenced by the data analysed. The authors have alluded to this briefly in the limitations section (lines 725-728) but not in specific detail.
Response: We thank the reviewer for this thoughtful comment. We agree that the TIMSS dataset does not include direct measures of executive functioning, which limits our ability to empirically model all cognitive pathways through which bullying may influence cognitive appraisals and mathematics anxiety, or learning outcomes. Our discussion of executive functioning was intended to provide theoretical context rather than to imply direct measurement. In response to the reviewer’s concern, we have revised this section and mentioned the necessity of further exploration of potential mediators such as executive functions, emotion regulation capacity, peer relationships and support.
At the same time, we would like to highlight that the contribution of the study lies in its cross-national estimation of the impact of bullying on mathematics anxiety, and domain-specific perceived control and value appraisals using propensity score matching that can yield causal estimates. The findings close an overlooked gap in bullying research and offer empirical evidence of how bullying relates to control-value theory (CVT) appraisals and mathematics anxiety across diverse cultural contexts. Also, Pekrun (2024) highlighted the need for the contribution of domain-specific studies to CVT. Accordingly, we have acknowledged in the limitations section that, although executive functioning was not measured, the study advances CVT and bullying research by demonstrating that exposure to bullying influence mathematics anxiety and appraisal-based pathways across countries. As stated in the manuscript, sensitivity analysis was conducted to measure potential direct effect of an omitted confounder, with moderate-to-high probability, on selection and outcome effect. Indirect effects of background cognitive mechanisms were not in the scope of the current study.
Comment 35:
Line 573: For clarity and comprehension, the authors could improve this sentence by adding that the stress they refer to arises from exposure to bullying.
Response: We thank the reviewer for this helpful suggestion. Accordingly, we have revised the sentence and added this for clarification: (e.g., stress stemming from exposure to bullying)
Comment 36:
Lines 601-602: Again, the study does not appear to have included the measurement of any physical or physiological aspect of mathematics anxiety. Rather a self-report questionnaire was the chosen instrument measuring this construct.
Response: As a response to Comment 27 all relevant terms including ‘physiological aspect of mathematics anxiety’ had been changed to ‘self-reported physiological aspect of mathematics anxiety’
Comment 37:
Lines 630-632: I believe this statement does not accurately reflect the current research findings on the status of bullying in Chilean schools / education system. According to national survey data from the Chilean ministry of education which measures bullying explicitly, Chile acknowledges bullying as a real and serious problem at the institutional, research and policy levels. Its denial is not systemic, rather it occurs more at the level of students’ perceptions or peer culture. It may be that certain behaviours which constitute bullying in some countries or regions may be minimised or not labelled bullying in others and therefore, they are under-reported rather than denied.
Response: We thank the reviewer for this important clarification. In response, we have revised the report and the sentence to more accurately reflect the Chilean context. The intent was not to imply institutional denial of bullying. The report, however, shows that many students who experience bullying do not report it to adults (teachers or parents), often because such behaviours are normalised or minimised among peers. This pattern is consistent with underreporting rather than systemic denial. The revised text now distinguishes clearly between institutional recognition of bullying and student-level underreporting driven by normalization and fear of retaliation. The revised statement: “In Chile, although bullying is formally acknowledged at policy-level, bullying is normalized among students; it is widely accepted as an inherent part of school life; and students rarely report bullying incidents to teachers or parents (Agencia de Calidad de la Educación, 2017)”
Comment 38:
Lines 643-644: I believe this to be somewhat of a blanket statement which does not accurately reflect the findings of the referenced report. According to this report, there is evidence that social and emotional skills education is on Chile’s agenda, that some teachers feel prepared and willing to teach these skills, and that some schools may already incorporate these practices. In fact, the referenced report provides an example of a targeted programme (Aulas en Paz: Peace education meets social and emotional skills in Chile, Colombia, p. 82-83) “that seeks to promote peaceful coexistence and reduce aggression in school environments, especially in vulnerable communities with histories of violence and substance abuse”. The authors even refer to this programme in lines 699-700. What the findings of the report do show is that there are large discrepancies in how these skills are promoted and in students’ own socio-emotional skills. Furthermore, the DOI of the referenced report listed in the bibliography list appears to be incorrect.
Response: Thank you for your comment. The OECD report shows that social-emotional learning is part of Chile’s agenda, and the Aulas en Paz program is one example. However, this program is for primary schools, not the middle-school context of our study. The report also highlights large discrepancies exist in students’ social emotional skills. We have rephrased our statement and mentioned ‘Peaceful Coexistence Campaign’ to better reflect the report’s nuanced findings.
“High level of awareness of bullying among school administrators in Chile, social-emotional skills education being on Chile’s agenda in accordance with ‘Peaceful Coexistence Campaign’, the normalization of school bullying, large discrepancies in students’ social and emotional skills in Chile (OECD, 2024b)”
We also checked the DOI (for OECD, 2024b), however, it works. Based on a comment of another reviewer, all references and DOIs were re-checked.
Comment 39:
Lines 703-706: A statement such as this requires a reference to the source of this information, otherwise it appears baseless.
Response: We thank the reviewer for this comment. The line numbers referenced did not correspond exactly to the submitted version or the version available in Susy platform, but we carefully examined the surrounding section and identified the sentence most consistent with the reviewer’s point. We have revised this sentence accordingly to address the reviewer’s concern: “In summary, the cited literature implies that psychological, sociocultural, economic, and state-level educational policy-related factors concurrently shape the degree of the effect of bullying on control and value appraisals.”
Comment 40:
Lines 770-773: This statement as written is conceptually risky and oversimplified, possibly even inappropriate, especially in a scientific report in the context of bullying, anxiety and educational outcomes. Bullying is not generally accepted in the scientific community as a genetic predisposition. Rather, as the authors previously state, peer dynamics, school climate, family factors and individual traits play a more significant role. The word “selfish” is also inappropriate. “Harmful” is perhaps a better choice here. As mentioned previously, the manuscript could benefit from professional language editing in order to avoid making unintentionally risky statements and enhance its impact.
Response: We thank the reviewer for highlighting this point and helping prevent a potential misunderstanding. The original sentence “it is necessary to recognize the evolutionary functions of bullying that are embedded in human genes” was intended to convey that, for some individuals, certain social outcomes of bullying may be reinforced within unhealthy relational dynamics, rather than suggesting that bullying itself reflects a genetic disposition. “it is necessary to recognize the evolutionary functions of bullying that are historically embedded in some human social contexts.” The term “selfish” in the Conclusion section was used rhetorically to underscore this idea, not as a scientific characterization (because Conclusion section is generally viewed as more interpretative and can accommodate a limited degree of normative reflection). This term would not be used in the Discussion section where interpretations are built on findings. Nonetheless, we appreciate the reviewer’s recommendation, and to be on the safe side, we have replaced the term with “harmful” to ensure greater clarity and to maintain a more reader-friendly tone.
Comment 41:
Lines 776-778: Although I agree with this statement, it contradicts the previously mentioned concept of bullying as genetically embedded. If bullying were primarily rooted in a genetic predisposition, eliminating or substantially reducing the behaviour through educational interventions would be extremely difficult.
Response:
Comment 42:
In conclusion, this manuscript addresses an important and timely topic and demonstrates commendable effort in both its theoretical framing and analytical approach. However, in my opinion, several areas require substantial revision before the work can be deemed publishable. The clarity and organisation of the introduction need improvement. Key constructs such as executive functioning and physiological aspects of anxiety are discussed but not measured, and several interpretations extend beyond what the data can actually support. Additionally, some claims, particularly those relating to the nature of bullying, need to be reconsidered to ensure they align with current empirical evidence and avoid conceptual overreach. With clearer theoretical grounding, more precise methodological reporting, and careful refinement of the discussion, the manuscript could make a valuable contribution to the literature on bullying and mathematics anxiety.
Response: We thank the reviewer for their thorough and constructive assessment of the manuscript. We appreciate the acknowledgement of the study’s relevance and the helpful guidance offered for improving its clarity and rigour. In response, we have revised the introduction for clearer organisation, refined the definitions of key constructs, and ensured that all interpretations remain aligned with the variables available in the TIMSS dataset. Claims related to bullying have been updated to reflect current empirical evidence, and methodological descriptions and the discussion section have been clarified accordingly.
We are grateful for these insights, which have strengthened the manuscript, and we hope the revised version addresses the reviewer’s concerns.

Reviewer 2 Report
Comments and Suggestions for Authors
Firstly, I would like to congratulate the author of this work for the excellent research carried out in preparing the manuscript, which represents a rigorous and valuable contribution to the study of academic emotions in cross-cultural contexts. The integration of Control-Value Theory with the phenomenon of bullying, as well as the use of advanced statistical methods on an international database, reflect meticulous, committed work that is highly relevant to the scientific and educational community. I sincerely commend your efforts in addressing such a sensitive issue as school bullying.
- TITLE AND ABSTRACT
Clear and concise, although it is recommended to end the abstract with a clear statement of implication/conclusion and avoid technical terms such as “distal antecedent” without a brief explanation.
- INTRODUCTION
Well-structured, solid justification of the cross-cultural approach, up-to-date and relevant literature review, relevance of the use of PSM compared to traditional correlational studies.
- THEORETICAL FRAMEWORK
The theoretical development is solid. The connection between bullying, control-value, and math anxiety is well-founded. It is suggested to briefly explore why bullying may affect “control” more than “value,” or vice versa, depending on the cultural context.
- METHOD
The design is appropriate (quasi-experimental with PSM), and the choice of countries is justified by cultural, socioeconomic, and educational criteria.
The supplementary file provides very detailed and rigorous information.
It is recommended to explicitly cite the tables in the supplementary file within the text. In the main text, justify more clearly the use of a single item for math anxiety. Although Wanous (1997) is cited, its weakness in capturing cognitive or behavioral components should be discussed.
- RESULTS
The results are presented clearly and concisely. I would recommend moving Figure 1 (PS distributions) to the supplementary material as it takes up a lot of space in the main text.
- DISCUSSION
The discussion interprets the findings in relation to CVT well. There is good integration between theory, empirical evidence, and context. The comparison and possible explanation of the variability between countries is clearly presented.
The discussion could be improved by including more explicit hypotheses about why Chile does not show a significant effect on control, or why it is only affected in certain contexts.
Additionally, the role of cultural factors (e.g., individualism, social norms, school climate) as hypothetical moderators could be considered in the discussion.
- CONCLUSIONS
The conclusions effectively summarize the key findings and mention practical and theoretical implications.
- REFERENCES
All bibliographic references added are up to date and relevant. However, some of them should be reviewed as they appear to be incomplete.
Author Response
TITLE AND ABSTRACT
Comment 1:
Clear and concise, although it is recommended to end the abstract with a clear statement of implication/conclusion and avoid technical terms such as “distal antecedent” without a brief explanation.
Response: We thank the reviewer for this helpful suggestion. In line with the recommendation, we revised the final sentence of the Abstract to present a clear implication of the findings: “Multi-country findings position bullying an antecedent of mathematics anxiety, highlighting the need for interventions grounded in psychological, sociocultural and educational policy factors to protect victims from the harmful effects of this behavior.”
Additionally, distal antecedent was briefly defined when it first appeared in the manuscript: “(broader factors that shape cognitive appraisals)”.
INTRODUCTION
Comment 2:
Well-structured, solid justification of the cross-cultural approach, up-to-date and relevant literature review, relevance of the use of PSM compared to traditional correlational studies.
Response: We thank the reviewer for this positive evaluation.
THEORETICAL FRAMEWORK
Comment 3:
The theoretical development is solid. The connection between bullying, control-value, and math anxiety is well-founded. It is suggested to briefly explore why bullying may affect “control” more than “value,” or vice versa, depending on the cultural context.
Response: We thank the reviewer for this helpful suggestion. In response, we have revised the manuscript to include a section (before Research Aim section) of how cultural contexts may shape the effect of bullying on cognitive appraisals and mathematics anxiety and whether bullying affects perceived control or perceived value more strongly.
METHOD
Comment 4:
The design is appropriate (quasi-experimental with PSM), and the choice of countries is justified by cultural, socioeconomic, and educational criteria.
Response: We appreciate the reviewer’s positive evaluation.
Comment 5:
The supplementary file provides very detailed and rigorous information.
It is recommended to explicitly cite the tables in the supplementary file within the text. In the main text, justify more clearly the use of a single item for math anxiety. Although Wanous (1997) is cited, its weakness in capturing cognitive or behavioral components should be discussed.
Response: We thank the reviewer for this helpful comment. Tables and Figures in Supplementary File were explicitly cited. We have now clarified in the manuscript why a single-item measure was used to assess mathematics anxiety. Specifically, large-scale assessments such as TIMSS employ single-item indicators for affective constructs to ensure cross-national comparability, and we provided more citations that demonstrates that single-item emotional measures show acceptable validity and strong correlations with multi-item scales (Wanous & Hudy, 2001; Gogol et al., 2014). We also acknowledge the limitation that such items do not fully capture cognitive or behavioral facets of anxiety.
RESULTS
Comment 6:
The results are presented clearly and concisely. I would recommend moving Figure 1 (PS distributions) to the supplementary material as it takes up a lot of space in the main text.
Response: Figure 1 was moved to the Supplementary File.
DISCUSSION
Comment 7:
The discussion interprets the findings in relation to CVT well. There is good integration between theory, empirical evidence, and context. The comparison and possible explanation of the variability between countries is clearly presented.
Response: We thank the reviewer for this positive evaluation.
Comment 8:
The discussion could be improved by including more explicit hypotheses about why Chile does not show a significant effect on control, or why it is only affected in certain contexts.
Response: We thank the reviewer for this insightful recommendation. There was a subsection discussing possible explanations specific to the Chilean context. Additionally, we clarified our interpretations with additional hypotheses and explanations about why Chile did not show a significant effect on perceived control.
Comment 9:
Additionally, the role of cultural factors (e.g., individualism, social norms, school climate) as hypothetical moderators could be considered in the discussion.
Response: We thank the reviewer for this valuable comment. In the revised Discussion, we added a brief reflection on potential cultural moderators such as individualism–collectivism. We also clarified that broad cultural dimensions alone do not fully account for the cross-national variation observed in our findings; instead, we note that psychological, sociocultural, economic, and state-level educational policy-related factors interactively provide a more comprehensive explanation for differences across countries.
CONCLUSIONS
Comment 10:
The conclusions effectively summarize the key findings and mention practical and theoretical implications.
Response: We appreciate the reviewer for this positive evaluation.
REFERENCES
Comment 11:
All bibliographic references added are up to date and relevant. However, some of them should be reviewed as they appear to be incomplete.
Response: We thank the reviewer for noting this issue. We carefully reviewed all references and corrected the entries that were incomplete to ensure accuracy and consistency with APA 7 and the journal’s guidelines.
Round 2
Reviewer 1 Report
Comments and Suggestions for Authors
Lines 320-330 The phrase “physiological symptoms of anxiety, i.e., nervousness” could be interpreted as somewhat reductive, since nervousness can also include cognitive or affective elements. You might slightly tighten this by emphasising self-reported nervousness as an indicator of physiological arousal.
Author Response
Comment 1:
Lines 320-330 The phrase “physiological symptoms of anxiety, i.e., nervousness” could be interpreted as somewhat reductive, since nervousness can also include cognitive or affective elements. You might slightly tighten this by emphasising self-reported nervousness as an indicator of physiological arousal.
Response: Thank you for this helpful suggestion. We have revised the statement as recommended to emphasise self-reported nervousness as an indicator of physiological arousal. Line 320-322:
“In light of the accessible and suitable data in the TIMSS 2019 dataset, the extent of this research is constrained to the self-reported nervousness as an indicator of physiological symptoms of anxiety, which individuals develop in response to mathematical tasks and outcomes in school.” ... "Therefore, the self-reported measurement of the physiological aspect of mathematics anxiety (specifically, self-reported nervousness, an indicator of physiological arousal) by a single item is justified. "
